

# Offline prompt reinforcement learning method based on feature extraction

Tianlei Yao, Xiliang Chen, Yi Yao, Weiye Huang and Zhaoyang Chen

College of Command and Control Engineering, Army Engineering University of PLA, Nanjing, China

## ABSTRACT

Recent studies have shown that combining Transformer and conditional strategies to deal with offline reinforcement learning can bring better results. However, in a conventional reinforcement learning scenario, the agent can receive a single frame of observations one by one according to its natural chronological sequence, but in Transformer, a series of observations are received at each step. Individual features cannot be extracted efficiently to make more accurate decisions, and it is still difficult to generalize effectively for data outside the distribution. We focus on the characteristic of few-shot learning in pre-trained models, and combine prompt learning to enhance the ability of real-time policy adjustment. By sampling the specific information in the offline dataset as trajectory samples, the task information is encoded to help the pre-trained model quickly understand the task characteristics and the sequence generation paradigm to quickly adapt to the downstream tasks. In order to understand the dependencies in the sequence more accurately, we also divide the fixed-size state information blocks in the input trajectory, extract the features of the segmented sub-blocks respectively, and finally encode the whole sequence into the GPT model to generate decisions more accurately. Experiments show that the proposed method achieves better performance than the baseline method in related tasks, can be generalized to new environments and tasks better, and effectively improves the stability and accuracy of agent decision making.

# INTRODUCTION

Reinforcement learning learns the optimal strategy by constantly interacting with the environment through trial and error, and this exploration process often results in high cost losses in real-world scenarios. For example, data sampling in the fields of healthcare and autonomous driving often poses certain risks (*Prudencio, Maximo & Colombini, 2022*). Offline reinforcement learning aims to learn from trajectories collected from a set of behavioral strategies, studying how to maximize the use of static offline datasets to train agents. Compared with standard reinforcement learning algorithms, offline reinforcement learning cannot directly interact with the environment to improve strategies, both learning objectives and strategy optimization are all centered around reward functions (*Zhang et al., 2023*).

Corresponding authors
Xiliang Chen,
lgd_chenxiliang@aeu.edu.cn
Yi Yao, lgd_yaoyi@aeu.edu.cn

The powerful sequence modeling ability of Transformer has shown great advantages in fields such as natural language processing. Based on this architecture, a series of generative pre-trained large language models such as GPT and BERT have had a significant impact on various aspects of real life (*Kalyan, Rajasekharan & Sangeetha, 2021*). The powerful text understanding and editing ability, image analysis ability and data mining ability of the model play an important role in many scenarios. In the field of reinforcement learning, Transformer has gradually been widely applied, especially in offline reinforcement learning. For example, Decision Transformer (DT) (*Chen et al., 2021*) integrate the trajectory of reinforcement learning (including states, actions, rewards, *etc.*) as a series of serial data into a Transformer-based pre-trained model to generate predicted actions, and model these sequence distributions to be the core task of learning. Such models can be trained using semi-supervised or self-supervised learning methods, which can effectively avoid the short-sighted behavior caused by reward discount and the instability caused by gradient signals in traditional reinforcement learning. Even using sophisticated policy improvement and value estimation methods, they show remarkably excellent performance in offline reinforcement learning. However, these methods share some of the same issues as offline reinforcement learning, as they rely entirely on static datasets during training and do not interact with the environment. This can lead to a decrease in accuracy when facing different data distributions, resulting in errors in practical applications (*Furuta, Matsuo & Gu, 2022*). Due to changes in the actual distribution, the model may not be able to find the best strategy in the test task and has some difficulty in generalizing and adapting preferences to previously unseen tasks. In addition, the Transformer is usually only able to receive a series of past observations, actions, and future returns, and cannot extract more detailed information from a single sequence, resulting in more accurate decisions (*Parisotto & Salakhutdinov, 2021*). Therefore, how to stably learn the optimal strategy in complex and diverse environments has become an urgent problem to be solved in such methods.

In order to further improve the stability and generalization ability of the model, based on the method of prompt learning, we extract prompt samples from the trajectories in the datasets, which contain information for task identification. The agent can quickly understand the representation information in the current task and learn the paradigm of learning sequence generation through these samples. And process the observation information in the input sample through both local and global methods to compare the results. First, only the observations in the prompt sample are segmented, and the processed sub-blocks are extracted through the long short-term memory (LSTM) layer for feature extraction, then the observation information in the whole input data is processed and features are extracted. Similar to the phrase sequence input in the field of natural language processing (NLP), we use the processed information to form a linear embedded sequence as the input of Transformer to model the joint distribution of trajectories, and generates actions to achieve the desired rewards through autoregressive methods, effectively improving the data utilization rate and decision-making accuracy, and can be more stable to generalize to different downstream tasks. Our code can be obtained at: https://github.com/YTL7/PLDT.git (https://doi.org/10.5281/zenodo.13558651). In summary, our main contributions are four-fold:

- We propose a novel method named Prompt Decision Transformer based on LSTM (PLDT), which transforms the traditional reinforcement learning problem into a sequence modeling problem, and uses GPT to predict the action autoregressively through advanced prompt learning and feature processing methods.
- We show that reinforcement learning can be well combined with prompt learning to better adapt to unknown tasks.
- We embed Vision Transformer (FiT) (*Dosovitskiy et al., 2021*) as the internal backbone of the model, and use LSTM for feature extraction, which effectively improves the accuracy.
- Through extensive evaluation of PLDT in environments such as Atari and OpenAI Gym, experiment results show that our method has superior performance compared to existing offline reinforcement learning (RL) algorithms.

## RELATED WORK

### Offline reinforcement learning

Reinforcement learning is a branch of machine learning that focuses on how to learn to make decisions, typically modeling the problem to be solved as Markov decision process (MDP). The ultimate goal of the decision-making process is to find a strategy in the set of policy functions that can maximize the expected cumulative discount return $R = \sum_{k=0}^{\infty} \gamma^k r^{k+1}$. The core difficulty of this type of method lies in its low sampling efficiency, which requires a large amount of interaction with the environment to collect training data, but this process will cause serious cost losses in most real-world scenarios. The core of offline reinforcement learning research is how to maximize the use of static datasets to train agents, and learn strategies by collecting static datasets from previously used unknown behavioral strategies (or multiple strategies). However, due to the lack of direct interaction with the environment, there may be distribution bias issues during the training process, leading to the learning of uncoordinated suboptimal strategies, resulting in overfitting or erroneous generalization (*McInroe, Albrecht & Storkey, 2023*).

*Kumar et al. (2020)* learned a conservative Q-function by adding a regularization term to the Q-value, generating a lower bound on the true value of the current strategy to eliminate the influence of partial extrapolation errors. *Xu et al. (2022a)* decoupled the profit maximization strategy in the traditional offline RL algorithm into a guidance strategy and an execution strategy, thereby achieving state concatenation of the datasets. *Yuan & Lu (2022)* designed a double-layer encoder structure that formalizes task representation learning by maximizing common information, reducing the impact of behavioral policy changes and better generalizing to non distributed behavior strategies. *Zhuang et al. (2023)* found that the Proximal Policy Optimization (PPO) algorithm can solve the offline RL problem without introducing any constraints or restrictions. By using the CLIP operation to achieve the constraint of the total variational distance between two policies, applying the constraint to the behavioral strategy can make the strategy learned by the PPO algorithm similar to the behavioral strategy, thus solving the offline problem.

## Decision transformer

Transformer has achieved excellent scores in fields such as NLP and computer vision (CV), and many researchers have also turned their attention to the field of reinforcement learning, especially in offline reinforcement learning. Decision Transformer (DT) regards reinforcement learning process as a sequence modeling problem. Different from traditional reinforcement learning modeling methods, DT uses Transformer to directly output actions for decision-making. Joint distribution modeling was carried out for sum of future reward $\hat{R}_t$ (return-to-go) (*Schmidhuber, 2019*), states $s$, actions $a$ and other trajectories in offline data. The trajectory after modeling is represented by Eq. (1), the subscript represents the time step.

$$\tau = (\hat{R}_1, s_1, a_1, \hat{R}_2, s_2, a_2, \dots \hat{R}_T, s_T, a_T). \tag{1}$$

DT inputs the encoded data information into the GPT model and predicts the action autoregressively through the causal self-attention mask (*Chen et al., 2020*). Compared with traditional reinforcement learning, the model avoids the short-sighted impact caused by discount reward and can adapt to multi-modal data.

*Gao et al. (2023)* enhanced DT by dynamic programming method, using advantage function instead of future reward. A separate neural network is introduced to approximate the function and use the learned value function to estimate the advantage. *Hu et al. (2023)* model the input sequence in offline reinforcement learning as a causal graph to predict actions and obtain the dependency relationship between different conceptual data, which to some extent alleviates the problem of long-term dependency learning caused by Transformer paying attention to all token information. *Wu, Wang & Hamaya (2023)* took a variable-length traversal trajectory as input to optimize the trajectory by keeping a longer history when the previous trajectory was optimal and a shorter history when it was suboptimal. Use expectation regression to train an approximation maximizer to estimate the highest attainable value for a given history length, then search for the history length associated with the highest value and use it for action inference. At present, most methods for sequential transformation based on Transformer are based on offline reinforcement learning (*Li et al., 2023b*). However, real-time reinforcement learning usually involves the online part and fine-tuning the pre-trained policy model on the offline datasets through interaction with the environment. *Zheng, Zhang & Grover (2022)* integrated offline pre-training and online fine-tuning in a unified framework, replaced the deterministic strategy in DT with the stochastic strategy, defined the trajectory level policy entropy to promote exploration during online fine-tuning, and combined the sequence regularizer with the autoregressive modeling target to realize efficient sample learning exploration and fine-tuning.

## Prompt learning

The essence of prompt learning is to unify downstream tasks into pre-training tasks, and transform downstream tasks into specific data forms through designed templates (*Gu et al., 2022*). The first application of prefix tuning (*Li & Liang, 2021*) is in the generation task. Different patterns are designed for different model structures. The goal of prefix

information is to guide the model to extract information related to x, where x refers to the raw data or information that is input into the model, so as to better generate output information y. It is often necessary to construct different prefix prompt templates for different model structures. For example, for the model of encoder–decoder architecture, both encoder and decoder need to add prefixes to get Formula (2):

$$z = [PREFIX1, X; PREFIX2, y]. \tag{2}$$

The purpose of adding prefixes on the encoder side is to guide the encoding of the input part, and the purpose of adding prefixes on the decoder side is to guide the generation of subsequent tokens. For the autoregressive architecture model, by adding a specific prefix to the input information, we get $z = [PREFIX; x, y]$. The appropriate preceding part of information can guide the generation of following part in the case of fixed large language model, such as contextual learning of GPT (*Brown & Mann, 2020*).

*Xu, Chai & Kordjamshidi (2023)* built a composite graph based on extracting the combination of objects and attributes from the training data, entered the updated concept representation into the prefix prompt to capture the composite structure, and learned it through the new prompt strategy. *Saito et al. (2022)* adopted the prefix prompt idea to solve the problem of large difference in the distribution of two kinds of data, by stitching two different prefix vectors in front of the two types of data to correspond to different information to help the model learn. *Lu et al. (2022)* perform prompt learning through policy gradient, in which the policy network selects the best context example from the candidate pool, and its optimization goal is to maximize the prediction reward of the given training example when interacting with the environment, thus maximizing the reasoning ability of the model.

## METHODOLOGY

In order to better enhance the few-shot learning ability and stability of the model to efficiently solve generalization problems, we proposed PLDT (Prompt Decision Transformer based on LSTM) based on the method of prompt learning. The overall framework of the model is shown in Fig. 1. Firstly, the trajectories in the datasets is sampled as prompt samples, and then the feature extractor is used to extract features from the data in the trajectories. In a prompt based framework, the prompt trajectories contained information for identifying tasks and entered in advance as a prefix. The model can transform the few sample generalization problem into a conditional sequence generation problem, and through this information, the model can quickly understand the representation information of some current tasks and the paradigm of sequence generation. In order to learn the information contained in the prompt samples and input trajectories more accurately, the observation sequences in the prompt samples and training samples are divided into specific sized sub-blocks, and the processed sub-blocks are feature extracted through the LSTM layer. Then, the processed information is combined into a linear embedding sequence and input into the model, transforming the RL problem into a conditional sequence modeling problem. The trained model can accurately understand

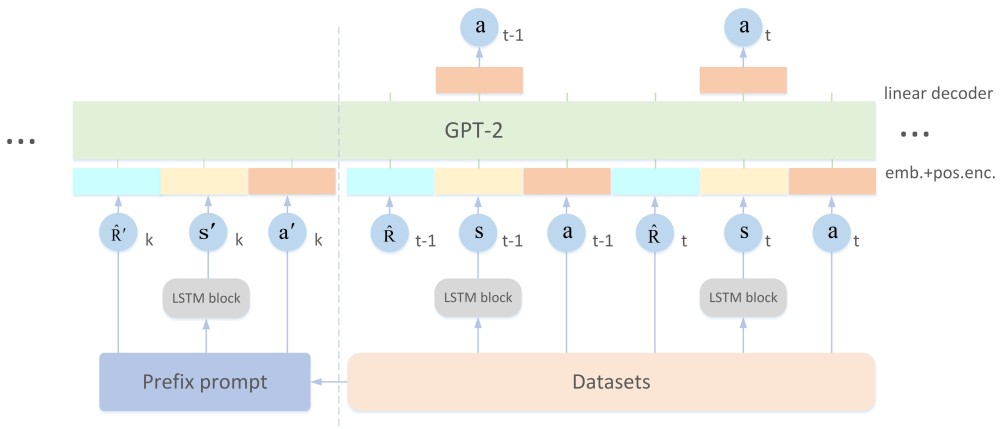

**Figure 1** **The architecture of PLDT.**

the semantics of the input sequence to find actions under expert level feedback, effectively improving the accuracy and stability of model decision-making, and better adapting to new tasks. Our code is available at: https://github.com/YTL7/PLDT.git, and the public dataset can be obtained at: https://rail.eecs.berkeley.edu/datasets/offline_rl/gym_mujoco_v2/.

## Construct prompt samples

Transformer can learn from few-shot samples when pre-training large-scale datasets, timely adjustment through prompt learning can make pre-trained models adapt to downstream tasks and more in line with human preferences (*Mayer, Ludwig & Brandt, 2022*). Prompt learning is widely used in NLP, by defining prompt templates, such as sentences, phrases, or keywords, it can help the model better understand input information. Finally, during prediction, the prompt template can be input into the model together with the data to be predicted. The principle of prefix tuning is shown in Fig. 2. However, due to the complex physical meaning and specific information of environment contained in the information of the RL trajectory, applying conventional prompt methods directly to RL tasks poses certain challenges, as RL prompts theoretically require modeling and analysis of the environment to guide the behavior of intelligent agents, rather than text descriptions or filling in missing information. This makes tweaking the prompt format for downstream tasks (as is the case in NLP) unlikely to yield significant improvements and has limited applicability.

We formalize the offline few-shot RL problem as a few-shot strategy generalization problem for new tasks. By distilling prompt samples from the original trajectory and adding them as prefix prompt before the input sequence, it is opposite to offline meta reinforcement learning that uses offline data or online interaction for specific tasks to update model weights (*Pong et al., 2022*). It achieves effective generalization without fine-tuning or gradient updates, to maintain high efficiency and avoid catastrophic forgetting caused by parameter changes. Specifically, in order to achieve few-shot learning in the context of RL, a subdataset $D'$ is distilled from the original datasets $D$, which is much smaller in size than the original datasets. The distillation method of the subdataset can be

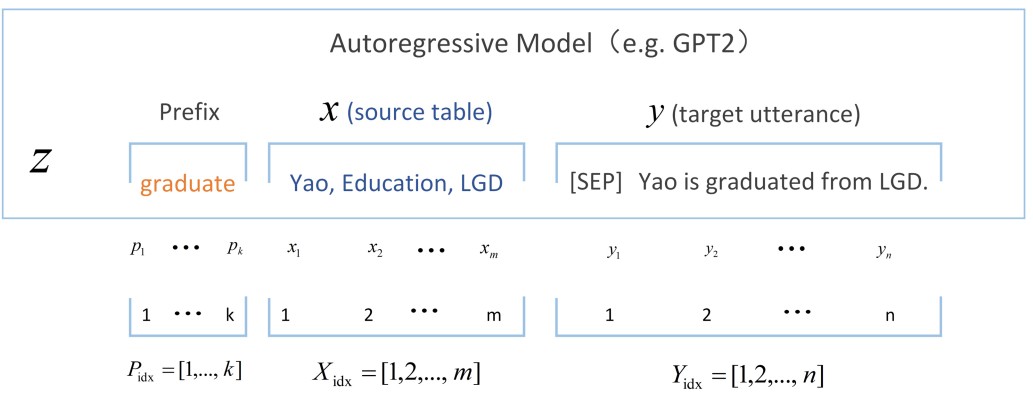

**Figure 2   Prefix tuning architecture.**

based on the use of gradient descent method for extraction, the random extraction method, or the reward value of each trajectory, as shown in Eq. (3), where $t$ represents the time step and $\tau$ represents the trajectory. We finally chose the first method to select the prompt samples through comparative experiments. These trajectories can more simply and directly reflect decision information under optimal strategies, including few-shot demonstrations for each task $Q_t \in Q_{train} \cup Q_{test}$ .

$$\tau'_t = max_{K'}[sort(\tau_t, R_t), \tau_t \in \tau]. \tag{3}$$

We designs an effective architecture that can directly extract unique task specific information stored in the demonstration datasets $D'$ and use this information to guide policy generation. Due to the presence of task information and data generation paradigms in the prompt samples, the model can quickly learn the dependency relationships in the sequence based on the prompt information. However, in general, text descriptions often require predefined language templates and may require significant labor costs (*Li et al., 2023a*). On RL tasks, using text or designing missing information templates similar to those used in NLP tasks to describe task information can have predictable high costs and uncertainties. In this paper, a sequence of trajectory segments in the reinforcement learning datasets with a length of $K'$ is used as a prompt sample, which $K'$ is much smaller than the length of any task $Q_t$. Therefore, prompt trajectory only contain the information needed to help identify the task, But it is not enough to provide all the information that the intelligent agent needs to learn. Utilizing the powerful sequence modeling capability of the Transformer architecture combined with prompt learning methods to more efficiently complete few-shot learning in offline RL (*Xu et al., 2022b*). Compared to text prompts in general NLP tasks, the difficulty of obtaining trajectory prompts is lower. Therefore, directly sampling trajectory segments $\tau'_t$ from the $D'$ constructed demonstration datasets as prompt samples can be trained without increasing additional costs. Each trajectory segment $\tau'_t$ in contains multiple states $s'$, actions $a'$, and reward values $r'$, $k$ represents the length of the input sequence, so prompt trajectory can store partial to complete sequence dependencies and data generation paradigms to specify tasks through implicit capture transformation

models and reward functions. Formally, the task $T_i$ defined trajectory prompts and the overall input trajectory are shown in Eqs. (4) and (5), respectively. By constructing prompt samples and utilizing a small amount of information, good results can be achieved, with strong few-shot learning ability.

$$\tau'_t = (r'_1, s'_1, a'_1, r'_2, s'_2, a'_2, \ldots, r'_{K'}, s'_{K'}, a'_{K'}) \tag{4}$$

$$\tau_{input} = (r'_1, s'_1, a'_1, r'_2, s'_2, a'_2 \ldots r'_{K'}, s'_{K'}, a'_{K'}, \hat{R}_1, s_1, a_1, \hat{R}_2, s_2, a_2 \ldots \hat{R}_K, s_K, a_K). \tag{5}$$

### Extraction of internal features

Compared with traditional deep neural network structure, Transformer lacks certain translation invariance and local perception, and it is difficult to achieve the same effect when the amount of data is insufficient. For example, using a medium-sized ImageNet trained Transformer will result in poorer accuracy compared to using a CNN structure (*Zhou et al., 2022*). LSTM structure is simple, easy to train and has good stability. It shows strong ability in feature extraction and can extract significant feature information from data, thus improving the accuracy and robustness of the algorithm (*Bhuvaneshwari et al., 2022*).

In the field of NLP, a sentence is usually divided into several tokens, which are then input into the Transformer. ViT (*Dosovitskiy et al., 2021*) applies the standard Transformer architecture directly to an image, splitting the entire image into small image blocks, and then sends the linear embedding sequence of the split image blocks as input to the Transformer network. This paper uses the idea of ViT to split the original observation (vector or image, such as four consecutive frames in an Atari game as one observation). By splitting the input dimension $H \times W \times C$ of the observation $s$ into a $N \times D$ matrix, these matrices are arranged into a $N \times (P^2 \times C)$ sequence, where $N$ is the length of the sequence and $D$ is the dimension of each vector. $C$ indicates the number of channels and meets the requirement $N = HW/P^2$. Get the sequence $\hat{o_p} = (o_p^1, o_p^2, o_p^3 \ldots o_p^N)$.

Existing research constructs a mapping from past data and values to actions by treating the reinforcement learning process as a series of trajectory sequences, models a mathematical expectation of the conditional probability of actions, and processes the historical trajectory $\tau = (\hat{R}_1, s_1, a_1, \hat{R}_2, s_2, a_2, \ldots \hat{R}_T, s_T, a_T)$ through Transformer. It includes environment state, agent state, history information, observation information and target information. As an effective feature extractor, LSTM can efficiently process various sequence data and extract various patterns and dependencies in the sequence, which plays an important role in natural language processing and time series analysis. Therefore, it is reasonable to use LSTM to extract features from $s$ in the trajectory. Current reinforcement learning efforts utilize Transformer to receive a state $s$ composed of past observations, We split the state $s$ into subblocks of a specific size, and each split information block is input into the LSTM neural network, as shown in Fig. 3. The output of the last time step is used as a feature representation for subsequent generation tasks. The LSTM output is shown in the following equation:

$$o_t = \sigma(W_o * [h_{t-1}, o_P^t] + b_o). \tag{6}$$

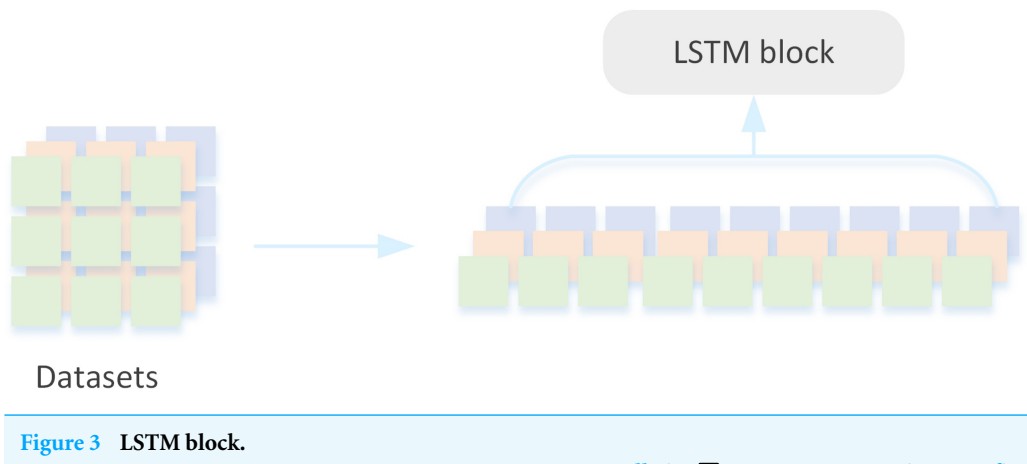

Datasets

**Figure 3  LSTM block.**

where $o_p^t$ is the inputs of the moment, $h_t$ is the hidden states, $W$ and $b$ are respectively the weight matrix and the bias term. Further, the processed information $o_t$ is converted to the embedding representation by Formula (7) and the location code is added.

$$o_e = [o_t^1 E; o_t^2 E; o_t^3 E; \dots o_t^N E] + E_{pos} \qquad E \in R^{(P^2 \cdot C) \times D}, E_{pos} \in R^{(N+1) \times D}. \tag{7}$$

Finally, $o_e$ with the return and action after coding are input into the set GPT model as a whole, and the information of future time steps is hidden, so that the model learns the semantics of the sequence and obtains the final predicted action according to the reward given by the posterior information.

**Specific processing of sequences**

When building the Transformer-based model, we focus on the last K time step input data, which contains a total of 3K tokens representing three different modes of return, state, and action, with additional prompt sequence of the same paradigm to enhance adaptability and generalization. In order to generate high-quality token embeddings from these raw inputs, the input data is mapped to preset embeddings dimensions by setting a linear layer. Then, layer normalization technique is used to stabilize these embeddings and improve the training efficiency and performance of the model. Standard Transformer can only accept one dimensional embedding sequence as input at each step. In order to be able to process 2D images and more specifically process individual observations, we do not simply input the status token directly into the linear layer by reshaping the input observations into a series of flattened 2D sub-blocks. Instead, a feature extractor is used to process the state data.

We didn't just apply the location information on the token embed. Instead, we learned an embed vector separately for each time step and added it to each token of the corresponding time step. This approach differs from positional embedding in the traditional Transformer model because here, a time step is actually associated with three different tokens (return, state, action). Ultimately, these processed tokens are fed into a GPT architecture model that uses autoregressive methods to predict future action tokens based on historical data.

In this way, our model can learn to extract useful information from historical data and make reasonable predictions based on it.

In general, PLDT reduces the error between the actions in the prompt samples and the historical predicted actions and the actions in the dataset, and can learn task-specific information from the data contained in the prompt trajectories. Combined with the historical data, the observations received at each step will be segmented using the idea of ViT, individual observation blocks will be processed through LSTM layer, and the Transformer will be used to process the overall sequential observation, finally the powerful generation ability of large models will be used to process sequences for future action prediction.

## Pseudocode

---

**Algorithm 1**

---

1: **Input:** offline datasets $D$, task demonstrations $P$, embedding layers $embed$, transformer with causal masking $GPT$, feature processor $InnerLSTM$, learning rate $\alpha$,

2: **def** $PLDT(r, s, a, t)$:

3:      sample trajectory $\tau$ of length $K$ from $D$

4:      sample the highest reward segment $\tau'$ of length $K'$ from $P$

5:      combine the two trajectory into $\tau^* = (\tau', \tau)$

6:      the observation $o_t$ in the $\tau^*$ is extracted by $InnerLSTM$, get $o_t^* = InnerLSTM(o_t)$

7:      $\tau_{input} = stack[embed(o_t^*, a_t, r_t) + E_{pos}]$

8:      hidden_state $= GPT(\tau_{input})$

9:      hidden_action $= unstack(\text{hidden\_state}).action$

10:      return $pred(\text{hidden\_action})$

11: **for** $(r, s, a, t)$ in dataloader **do**

12:      *Prediction action* : $a_{pred} = PLDT(r, s, a, t)$

13:      *Loss function* : $loss = \sum(a - a_{pred})^2$

14:      $\theta = \theta - \alpha \bigtriangledown_\theta loss$

15: **end for**

---

# EXPERIMENT

## Environments

D4RL (*Fu et al., 2020*) is a large and public dataset created by Google for evaluating reinforcement learning algorithms. The dataset is based on real-world tasks, including the recording of intelligent agents in interactive environments (such as autonomous driving Carla, AntMaze, Mujoco, *etc.*), has simple and complex classifications with a wide variety of types, aiming to test and compare the performance of different reinforcement learning algorithms in real-world environments.

To verify the effectiveness of the proposed algorithm, three continuous control tasks in the D4RL benchmark are selected firstly, which require fine-grained continuous control,

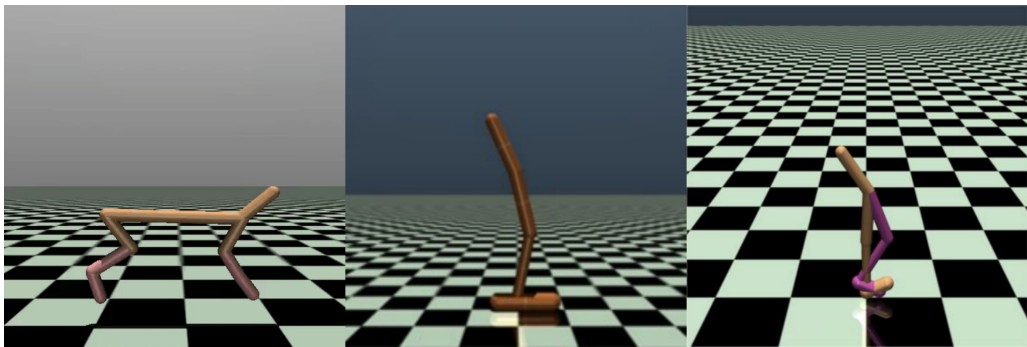

**Figure 4** MuJoCo environment.

**Table 1** Information of environments.

| Environment | State dimension | Action dimension | Sample size |
| --- | --- | --- | --- |
| HalfCheetah | 17 | 6 | $10^6$ |
| Hopper | 11 | 3 | $10^6$ |
| Walker2d | 17 | 6 | $10^6$ |
| Cheetah-dir | 17 | 6 | $10^6$ |

namely HalfCheetah, Hopper and Walker2d under MuJoCo. As shown in Fig. 4, the HalfCheetah environment simulates a robot with two legs that aims to run quickly on a level surface by controlling the angle and speed of its joints. The Hopper environment simulates the walking motion of a one-legged robot, which aims to complete a variety of tasks while walking steadily by controlling the angle and speed of its joints. The Walker environment provides a realistic bipedal robot walking simulation environment, and its goal is to make the robot move as fast as possible. By observing these signals, the walking performance of the robot can be evaluated, such as walking speed, stability, energy consumption, *etc*. The specific parameters of environments are described in Table 1.

## Details

In this section, the effectiveness of the proposed method is verified by training in the MuJoCo environment task. Firstly, we extract the features of the observation information on the Cheetah-dir task by both local and global input methods and compare the effects. Through the second more effective processing method, the rewards obtained by PLDT method and baseline method in different tasks were compared on two datasets with medium expert and medium datasets respectively. In addition, the normalized score performance of the method proposed was compared with classic reinforcement learning algorithms and currently more advanced algorithms in the field. Then the effectiveness of prompt trajectory learning and feature processing of observation information in this paper was verified through ablation experiments.

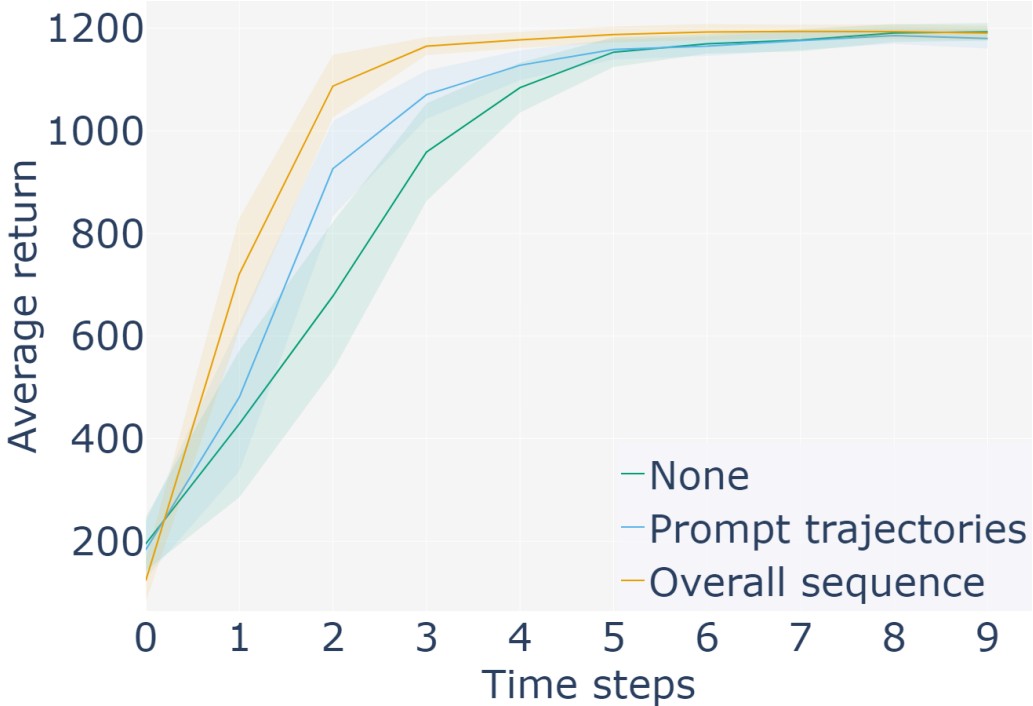

**Figure 5** Compare the feature processing of global and local sequences.

## Experimental results and analysis

This section will present the experimental results of PLDT algorithm and baseline algorithm in various tasks on the D4RL datasets, including the medium expert datasets and the medium datasets. The experimental results ultimately meet the convergence conditions, and different random seeds are used for multiple experiments to calculate the mean and obtain the final experimental data. The experimental results will be analyzed.

## Benchmark comparisons

In order to verify that PLDT method can achieve better performance, the Cheetah-dir task first compared the effects of feature extraction only on the observation information in the prompt samples and on the observation information in the overall input samples, with the addition of prompt trajectories. As shown in the Fig. 5.

Figure 5 shows that the two methods have different amplitude of improvement compared with the baseline DT method, and the effect of processing the overall input observation information is more significant. Further, through the second and more effective processing method, comparative experiments were carried out in three continuous control tasks such as HalfCheetah, Hopper and Walker2d in MuJoCo environment. Two different offline datasets settings were used to conduct the experiment, including the medium and medium-expert datasets. The medium datasets is generated by collecting standard reinforcement learning algorithms (SAC) for online training strategies, then stopping training and collecting a specific number of samples from the trained strategies. The

medium expert datasets is generated by mixing different numbers of expert demonstration sets and suboptimal data. To maintain consistency, the hyperparameters in PLDT method and DT method are identical except for the improved method. The experimental results were shown as follows.The yellow line represents the score obtained by adding the method in this article, and the blue line represents the score obtained by the baseline method.

## Baselines

We compared the proposed PLDT method with six benchmark methods, including classical offline RL algorithms as well as state-of-the-art algorithms.

**DT**: Decision transformer (*Chen et al., 2021*) adopts a pure supervised learning approach. It uses Transformer to directly output actions for decision-making, and trains the model by maximizing actions under given historical trajectory and future return conditions.

**BC**: Behavior clone (*Wu, Tucker & Nachum, 2019*) algorithm turns the imitation learning problem into a supervised learning problem, learns a strategy network through a series of binary sets (states, actions) in the expert data set, and then predicts the corresponding action according to the input state, making the predicted action as close as possible to the action of the human expert.

**CQL**: Conservative Q-learning (*Kumar et al., 2020*) learns a conservative lower bound of the Q function by adding regular terms to the Q function, reducing excessive exploration of the policy in the unexplored region, thereby avoiding overestimation of the Q value, and improving the stability and security of the policy.

**BRAC**: Behavior Regularized Actor Critic (*Wu, Tucker & Nachum, 2019*) algorithm is a behavior regularized offline reinforcement learning algorithm, aims to solve the problem of data distribution offset in offline data sets. The core idea of algorithm is to regularize the learning policy by introducing the distribution information of the behavior policy, and to avoid over-fitting to incorrect data distribution in offline environment.

**StAR**: StARformer (*Shang et al., 2022*) explicitly models the short-run state-action-reward representation (STAR-representation), introducing Markov-like inductive bias to improve the long-run modeling. Extract the StAR representation by self-focusing image state blocks, actions, and reward tokens within a short time window, and then combine it with a pure image state representation (extracted as convolutional features) to self-focus the entire sequence.

**GDT**: Graph Decision Transformer (*Hu et al., 2023*) models input sequences as causal graphs to better capture potential dependencies between adjacent states, actions, and rewards, and to distinguish the effects of these different tokens. Fine-grained spatial information can be accurately collected and integrated into motion prediction, which improves performance.

As shown in the Figs. 6 and 7, the PLDT method proposed in this paper achieved better results than the baseline DT method in three tasks under two different level setting datasets. As shown in Fig. 6A in the HalfCheetah task, the PLDT method achieved significant results compared to the DT method, receiving rewards far higher than the baseline method, reaching the convergence interval at time step 60, and achieving a high-performance score

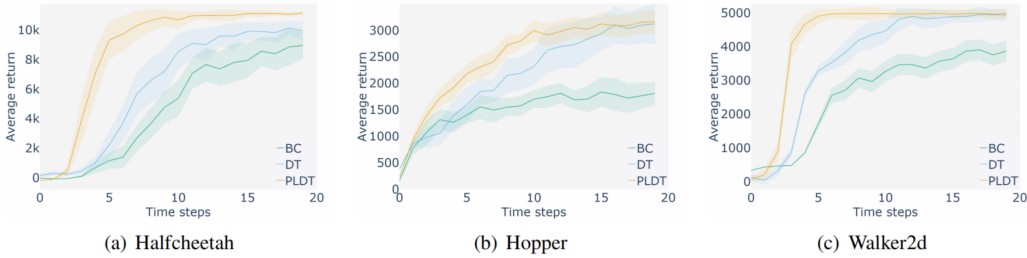

**Figure 6** Comparative experiments with baseline methods on medium expert datasets.

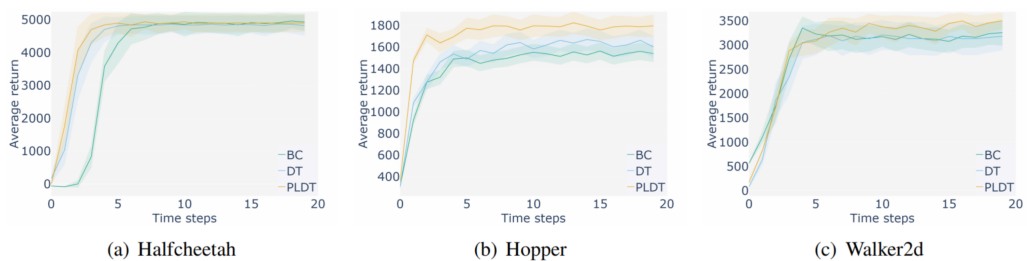

**Figure 7** Comparative experiments with baseline methods on medium datasets.

close to 12000. As shown in Fig. 6B, in the Hopper environment, the PLDT method and DT method achieved scores that were very close at the beginning of training. In subsequent training, their advantages gradually emerged and they achieved better results, and the overall score level remained ahead. As shown in Fig. 6C, the PLDT method achieved faster convergence speed and higher scores in the Walker2d task. It had already achieved a score close to the highest near time step 30 and remained within that range throughout the subsequent process. Figure 7 shows the experiment conducted on a medium datasets. Due to the lower overall quality of trajectory prompt samples compared to the medium expert datasets, the stability and performance comparison of the algorithm decreased on the medium expert datasets, but still achieved better performance than the baseline DT method. In addition, Table 2 includes more powerful baseline methods for comparison and normalizes scores based on the methods established in D4RL, where bold data indicates the highest score achieved for the same task. The comparative data includes GDT based on DT improvement, CQL (*Kumar et al., 2020*) and BRAC (*Wu, Tucker & Nachum, 2019*) based on model free, as well as several imitation learning algorithms, including BC and StARformer (*Shang et al., 2022*). The comparative data comes from experimental results in various original papers (*Chen et al., 2021*; *Hu et al., 2023*; *Fu et al., 2020*). Through the comparison in the table, our method achieved better performance in most tasks, indicating that the PLDT has significant competitiveness.

**Influence of prompt quality:** In order to further verify the influence of prompt trajectory quality on model performance, medium expert, medium and random prompts were used to conduct experiments on random, medium and medium expert datasets, respectively.

**Table 2  Results for D4RL datasets.**

| Dataset | Environment | PLDT | DT | CQL | BRAC | BC | StAR | GDT |
|---|---|---|---|---|---|---|---|---|
| Medium-Expert | HalfCheetah | 94.1 ± 0.8 | 86.8 ± 1.3 | 62.4 | 41.9 | 59.9 | 93.7 | 92.4 ± 0.1 |
| Medium-Expert | Hopper | 110.8 ± 1.4 | 107.6 ± 1.8 | 111.0 | 0.8 | 79.6 | 111.1 | 111.1 ± 0.1 |
| Medium-Expert | Walker2d | 112.6 ± 0.1 | 108.1 ± 0.2 | 98.7 | 81.6 | 36.6 | 109.3 | 107.7 ± 0.1 |
| Medium | HalfCheetah | 43.4 ± 0.2 | 42.6 ± 0.1 | 44.4 | 46.3 | 43.1 | 42.9 | 42.9 ± 0.1 |
| Medium | Hopper | 69.4 ± 1.1 | 67.6 ± 1.0 | 58.0 | 31.1 | 63.9 | 59.5 | 65.8 ± 5.8 |
| Medium | Walker2d | 75.4 ± 0.9 | 74.0 ± 1.4 | 79.2 | 81.1 | 77.3 | 73.8 | 77.8 ± 0.4 |
| | Average | 84.3 | 81.1 | 75.6 | 47.1 | 60.1 | 81.6 | 83.2 |

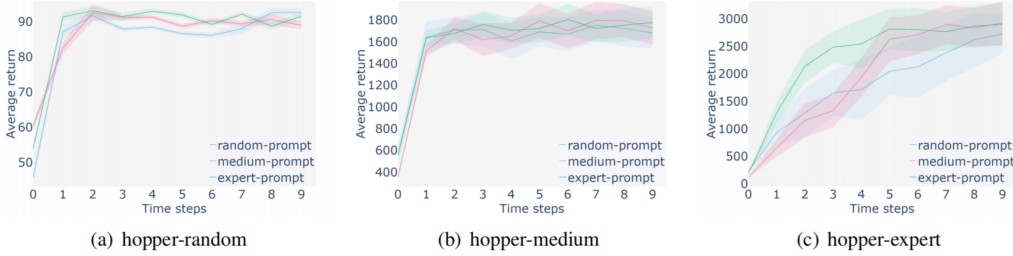

(a) hopper-random          (b) hopper-medium          (c) hopper-expert

**Figure 8  The influence of the quality of the prompt trajectory on the experimental results under different levels of datasets.**

As shown in Fig. 8, trajectories of three different qualities are respectively used as prompt samples on three levels of datasets under Hopper task, and the results show that the medium expert prompt samples can better guide the model to make decisions with higher rewards in general.

**Performance in delayed reward setting:** TD-based reinforcement learning requires intensive rewards to get better scores, so that the agent can get immediate feedback on the decision at every time step, thus speeding up the learning speed and improving the accuracy. However, the ideal sufficient reward signal is difficult to achieve in reality. This paper considers the delayed return setting of the D4RL benchmark, in which the agent receives no feedback midway through training and only receives the cumulative reward of the trajectory in the final time step. As shown in Fig. 9, experiments under delayed reward settings show that the proposed method is more robust to sparse reward environments.

**Performance in sparse reward environment:** Maze2D is a navigation task focused on evaluating the agent's trajectory stitching ability. The core of Maze2D is not simply to find the shortest or optimal path from the starting point to the end point, but to focus on how the agent can effectively use and splice existing sub-optimal trajectory fragments to build a complete path that can successfully reach the target point. In this environment, we tested the trajectory splicing ability of PLDT method under sparse reward conditions. The results are shown in the Fig. 10, the method proposed in this paper performs better and has more brilliant stability under four maps with different settings.

**Ablation study:** In order to demonstrate the effectiveness of the methods proposed in this algorithm, such as adding trajectory prompts and feature processing of observation

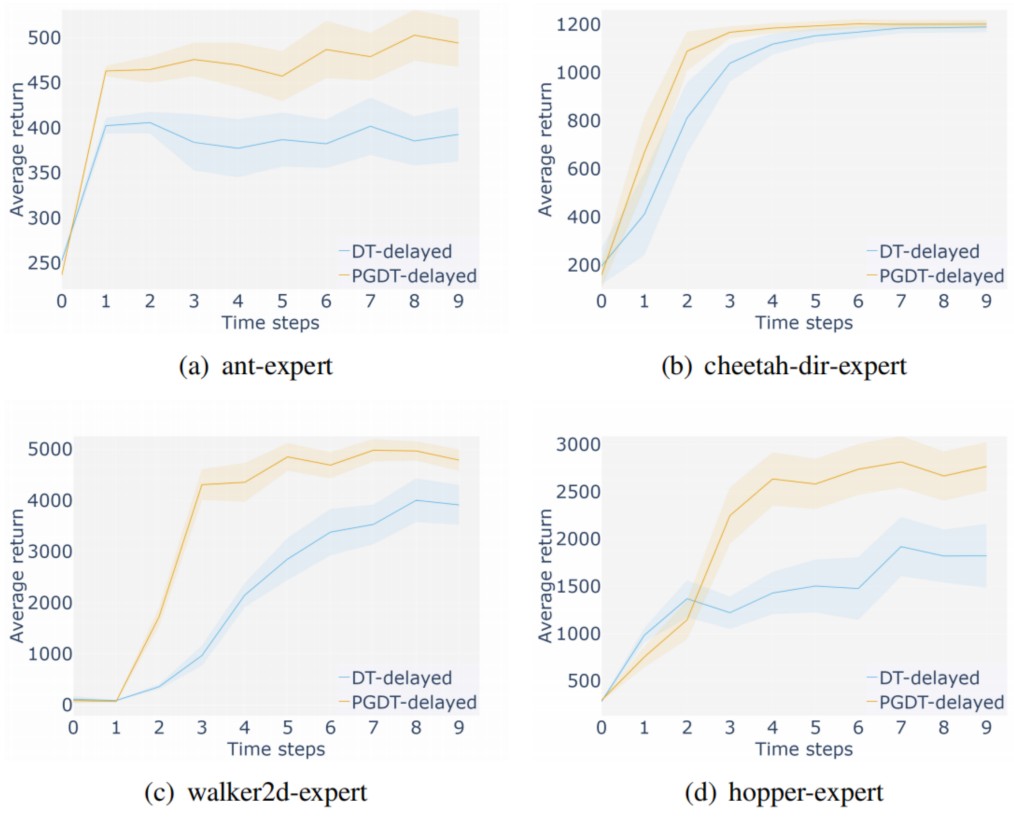

**Figure 9 Experimental results under the delayed reward setting.**

information, ablation experiments were analyzed in this section. We selected medium expert and medium datasets in a representative Hopper environment for experiments and compared them with the baseline DT method. The results are shown in the following figure.

Figures 11 and 12 show the ablation experiments conducted using two different level datasets in the Hopper environment, where the yellow line represents the score obtained by incorporating the method proposed in this paper, and the blue line represents the score obtained by the baseline DT method. Figure 11 represents the comparison between adding prompt trajectories and not adding prompt trajectories on two levels of datasets. The results show that after adding prompt samples, the model can adapt to the task faster to reach the convergence interval and achieve better scores, which is better than the baseline method. Figure 12 represents a comparison between the results of segmenting and extracting features from the observation information in the input sequence and not processing that information. The experiment shows that the former can help the model learn more efficiently, achieve higher performance scores, and achieve better stability. The above ablation experiments have demonstrated the effectiveness of the proposed method, which not only enhances the stability of the algorithm but also achieves higher average returns.

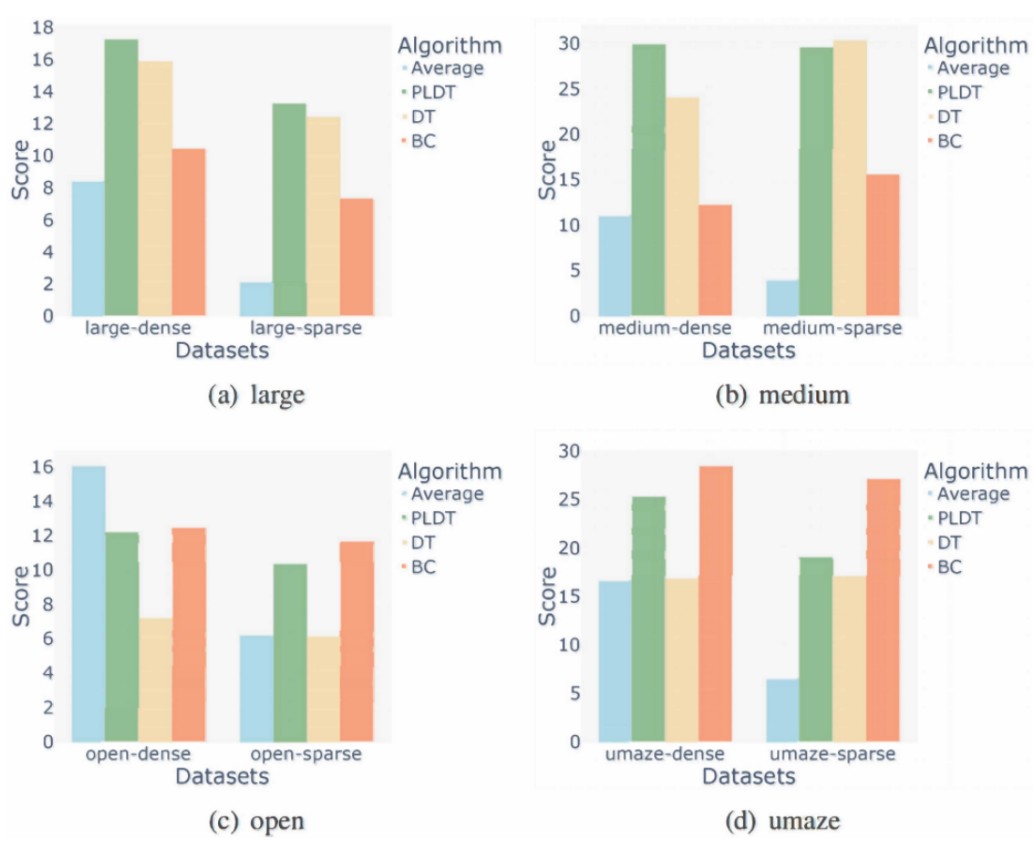

**Figure 10 Experimental results from four settings of the map under the Maze2d task.**

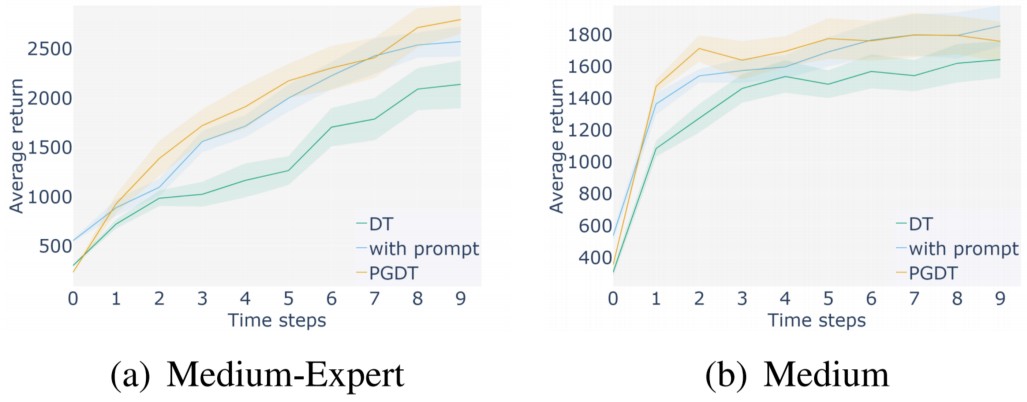

**Figure 11 Ablation experiments using only prompt learning.**

## DISCUSSION

### Why does PLDT perform well?

In order to solve the problem of high dimensional state space and action space, the traditional reinforcement learning uses neural network to fit the value function or strategy

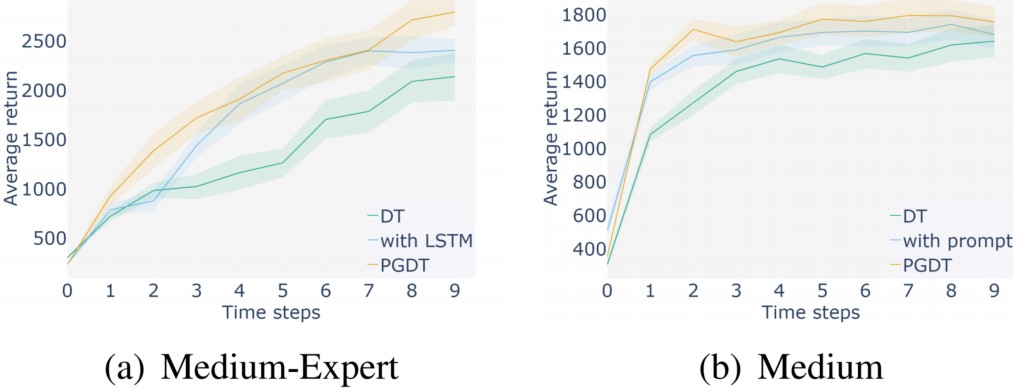

(a) Medium-Expert          (b) Medium

**Figure 12** Ablation experiments using only state features processing.

function. At present, most of the research is based on the improvement of value function or strategy, but from the perspective of sequence data modeling provides a new idea for the study of deep reinforcement learning. We use transformer to model the combined modeling of state, action, and return sequences, encoding the modeled trajectories through the processing layer. At the same time, the large language model can learn from fewer samples, combined with the method of prompt learning and supervised learning, so that the model can quickly adapt to the current task and imitate the given prompt trajectory. In order to better process sequence information, we also use the idea of ViT to slice and encode a given observation, use the internal LSTM layer for feature extraction, and finally input the output of this layer and the modeling trajectory into the external transformer model to generate new strategies.

Given the initial state condition, our proposed method can accomplish the strategy enhancement according to the highest possible return. Compared with traditional off-line reinforcement learning methods, PLDT abandons Markov property and does not use any constraints on value functions or strategies, avoiding the classic fatal triangle problem in reinforcement learning. In general, it benefits from the following points:

1. Compared with mainstream offline RL algorithms, by using sequences to model targets and training the model according to the collected experience, the long-term bootstrap process can be avoided. By making assumptions about expected rewards in the future, you can avoid the short-sightedness associated with reward discounts

2. Transformer can use the self-attention mechanism directly to complete credit-assignment for scenarios with sparse rewards. And can model a wide distribution, suitable for generalization and migration

3. By combining the prompt learning, the model can have stronger generalization ability.

4. Combining the structure of ViT with the LSTM network to process inputs can better capture the features of individual observations to improve accuracy and performance.

### Deficiencies and limitations

Although PLDT can effectively increase generalization and stability, depending on prompt learning makes the model have certain requirements on the quality of prompt trajectories, and high-quality trajectories can guide the model to achieve better results. However, it is often difficult to obtain high-quality prompt trajectories. In the future, more reasonable and effective prompt samples can be used to help construct better prompt learning methods.

In a short time step, due to the uncertainty of the random prompt trajectories, the trajectories with poor quality may be matched as the prompt samples, resulting in low learning efficiency of the model, but it will quickly improve with the increase of time steps. Therefore, how to better select prompt samples or combine with more efficient prompt learning methods can be a research point in the future.

In addition, as shown in the experiment, PLDT takes more time in calculation than baseline method, and how to balance calculation cost and performance needs to be further studied.

### Future application

Due to its efficient sequence dependency capturing capability and powerful modeling capability, PLDT can process decision sequences of different lengths without the need for a complete environment model. Instead, it can conduct efficient training through historical decision data to learn hidden patterns and generate high-quality new decisions, which is suitable for a variety of application scenarios. Such as game strategy, resource allocation and path planning. Through the efficient sampling capability of PLDT, it can stably learn from online or offline multi-modal data, which brings more possibilities in multi-agent reinforcement learning, and can be combined with a variety of algorithms, which can be used as a preliminary strategy or for policy improvement.

## CONCLUSION

We propose the PLDT method which uses the represented trajectories in the datasets as few-shot prompt. During each training session, We first sample trajectories in the task set as prompt samples to guide the model to adapt more quickly to unseen tasks. The features of observation information are then extracted from the prompts and training trajectories to obtain more accurate dependency relationships. The experimental results demonstrate the effectiveness of the proposed method, which have certain advantages over existing offline reinforcement learning algorithms. Our method exhibits robust performance on datasets containing sub-optimal trajectories and can be extended to out of distribution generalization environments. And it was verified that the quality of sample prompts has a significant impact on the performance of the model, expert level sample prompts can significantly improve the accuracy and stability of model predictions.

With the development of new technologies, future research can use more reasonable and effective prompt samples to assist in building better prompt learning methods, design appropriate objective functions to balance the weight of trajectory prompts and historical context to adapt to more diverse environments and solve more complex tasks. The current methods mainly capture dependency relationships in decision sequences through

**Table 3   Hyperparameters of PLDT for MuJoCo and Adroit experiments.**

| Hyperparameters | Value |
| --- | --- |
| Prompt length | 5 |
| Context length K | 20 |
| Dropout | 0.1 |
| Learning rate | 1e−4 |
| Weight decay | 1e−4 |
| Return-to-go condition | 3600 Hopper |
| | 5000 Walker |
| | 12000 HalfCheetah |
| Embedding dimension | 128 |
| Batch size | 64 |
| Attention heads | 1 |
| Inner blocks | 1 |

**Table 4   Hyperparameters of PLDT for Atari experiments.**

| Hyperparameters | Value |
| --- | --- |
| Prompt length | 5 |
| Context length K | 50 Pong |
| | 30 Breakout |
| | 30 Qbert |
| | 30 Seaquest |
| Dropout | 0.1 |
| Learning rate | 6 * 1e−4 |
| Weight decay | 1e−4 |
| Return-to-go condition | 120 Breakout |
| | 5000 Qbert |
| | 20 Pong |
| | 1450 Seaquest |
| Embedding dimension | 64 |
| Batch size | 64 |
| Attention heads | 1 |
| Inner blocks | 1 |

Transformers and abstract strategies, which mainly rely on the support of a large number of offline datasets. However, it is not feasible for some decision-making tasks to break free from online frameworks in practical applications, so how to transfer models to online decision-making tasks through special paradigm designs is also a problem that needs to be solved in the future.

## APPENDIX A. HYPERPARAMETERS

We show the hyperparameters of PLDT in Tables 3 and 4.

**Table 5** Results for Atari datasets.

| Game | CQL | BC | StAR | QR-DQN | DT | GDT | PLDT |
|------|-----|-----|------|--------|-----|-----|------|
| Breakout | 211.1 | 138.9.1± 61.7 | 436.1 ± 63.6 | 17.1 | 267.5 ± 97.5 | 393.5 ± 98.8 | 414 ± 77.2 |
| Qbert | 104.2 | 17.3.1 ± 14.7 | 51.2 ± 11.5 | 0 | 15.4 ± 11.4 | 45.5 ± 14.6 | 47.8 ± 8.9 |
| Pong | 111.9 | 85.2.1 ± 20.0 | 110.8 ± 60.3 | 18 | 106.1 ± 8.1 | 108.4 ± 4.7 | 117 ± 15.3 |
| Seaquest | 1.7 | 2.1 ± 0.3 | 1.7 ± 0.3 | 0.4 | 2.5 ± 0.4 | 2.8 ± 0.1 | 3.1 ± 0.1 |

## APPENDIX B. SUPPLEMENTARY EXPERIMENTS

To test our method on discrete action tasks, we chose the Atari game as the experimental setting. The Atari is a game console released by Atari in October 1977, featuring classic games such as Pong, Breakout, and Qbert. We evaluated our approach on 1% of samples from the DQN-Replay dataset, which collected 500,000 of the 50 million migrations observed by online DQN during training for Atari tasks. The dataset is public and can be obtained from https://console.cloud.google.com/storage/browser/atari-replay-dataset.

We compared the current advanced algorithms on four Atari tasks (Breakout, Qbert, Pong, and Seaquest). We used three different random seeds and averaged the results, Table 5 shows the mean and variance of standardized scores for the experiment, with the best results highlighted in bold, the experimental results show that PLDT method can also achieve relatively better results in discrete tasks. As for the experimental results of the baseline algorithm, we obtained them directly from the relevant papers (*Chen et al., 2021*; *Hu et al., 2023*; *Fu et al., 2020*).

The Adroit Robotic Arm dataset is a dataset for the study of robotic arm operation, containing data from a variety of sensors, involving the control of a 24-DoF simulated Shadow Hand robot, which is capable of performing tasks such as hammering nails, opening doors and rotating pens. The tasks were chosen to measure the impact of narrow expert data distributions and human demonstrations on sparsely rewarded high-dimensional robotic manipulative tasks (*Fu et al., 2020*). We compared the proposed method with the baseline methods, we used three different random seeds and averaged the experimental datas, the results are shown in the Fig. 13. In all the tasks in this environment, our method enabled the robotic arm to take relatively optimal actions and achieved best scores. The dataset is public and available at https://sites.google.com/view/deeprl-dexterous-manipulation.

## APPENDIX C. SUPPLEMENTARY ABLATION EXPERIMENTS

We compared the effects of different prompt lengths on performance, which was set as 5, 10, 15, and 20 respectively. We performed ablation experiments on the pen task in Adroit, the experimental results are shown in Fig. 14. The result showed that increasing the length did not demonstrably improve the score, and still achieved excellent results under the default length of 5.

For how to choose the prompt trajectory, we compared the two methods of direct select of the highest reward prompt trajectory and random selection. We performed ablation experiments on expert dataset of Walker2d and Hopper, the experimental results are shown

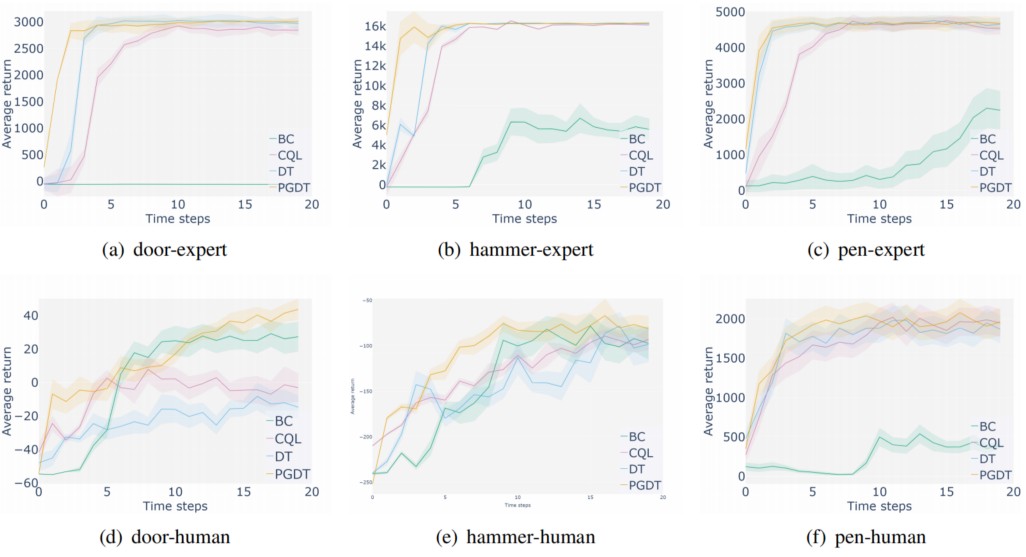

**Figure 13** **Comparative experiments with baseline methods.**

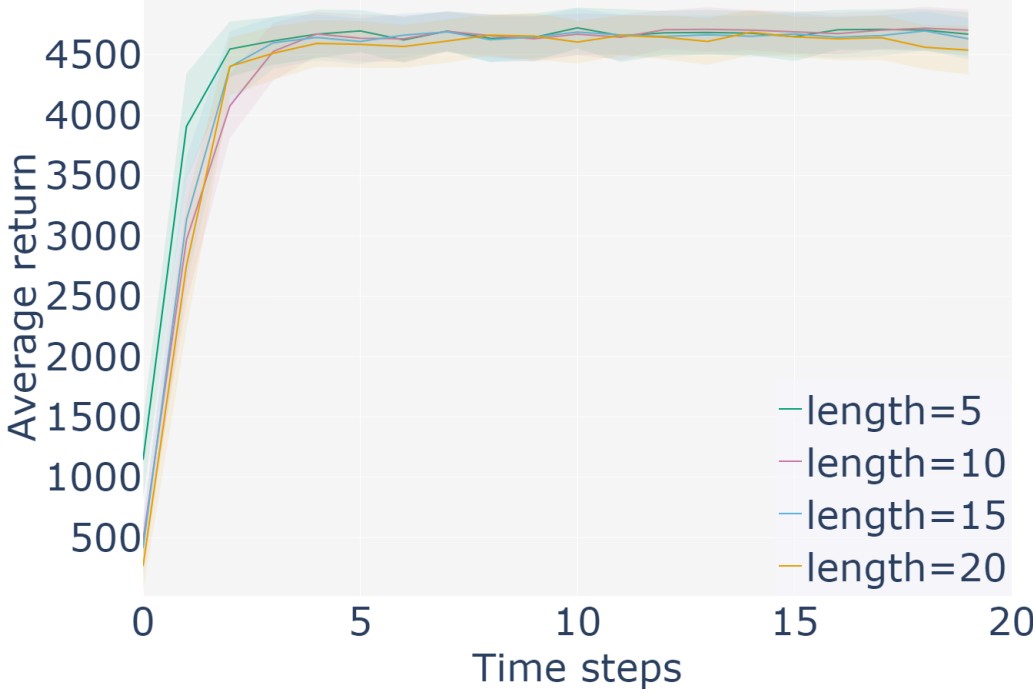

**Figure 14** **Ablation study on the length of prompt trajectory.**

in Fig. 15. Experiments show that random selection of prompt samples can achieve better scores and cover a wider range of task information, while direct use of the highest reward trajectory as prompt samples has the theoretical risk of overfitting.

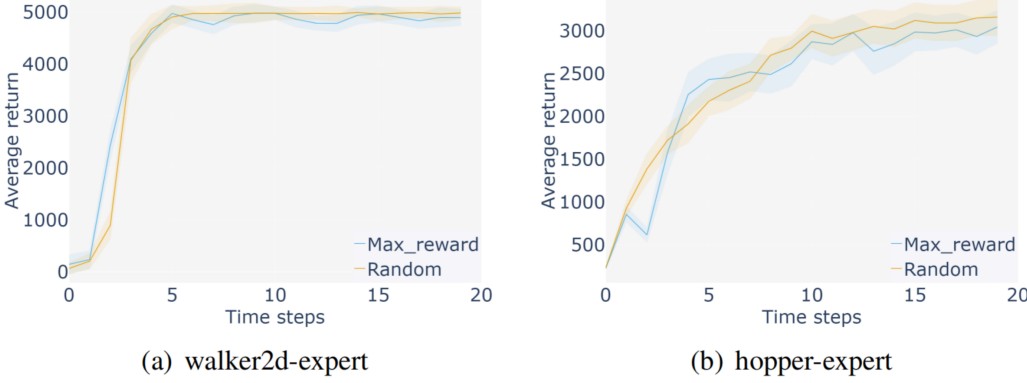

(a) walker2d-expert          (b) hopper-expert

**Figure 15** Ablation study on the selection method of prompt trajectory.

**Table 6  Configuration environment.**

| Name | Value |
| --- | --- |
| Operating system | Windows 10 Version 22H2 |
| Development language | Python 3.10 |
| Frame | Pytorch 2.0.0 + Cuda 11.7 |
| CPU | Intel(R) Xeon(R) Gold 6226R |
| GPU | NVIDIA RTX 5000 |
| GRAM | 48GB |
| RAM | 64GB |
| HDD | 1TB |

**Table 7  Comparison of computing resource occupancy.**

| Name | CQL | BC | DT | GDT | PLDT |
| --- | --- | --- | --- | --- | --- |
| CPU | 27% | 24% | 29% | 33% | 32% |
| GPU | 17% | 12% | 19% | 22% | 23% |
| RAM | 18% | 16% | 18% | 20% | 19% |

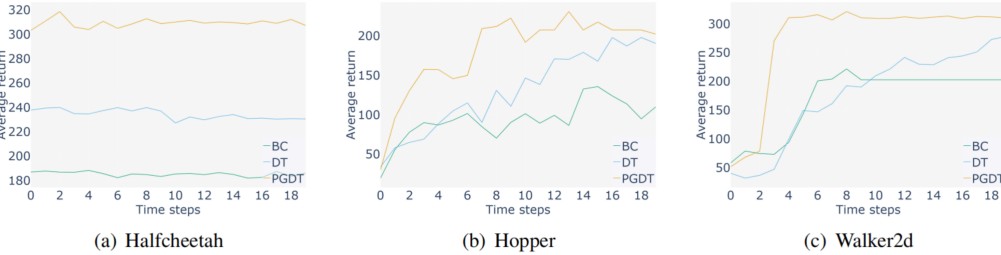

(a) Halfcheetah        (b) Hopper        (c) Walker2d

**Figure 16** **Comparison of running time.**

# APPENDIX D. DETAILED RESULTS

We describe the environment configuration of the experiment in Table 6, and compare the resource usage with other algorithms in Table 7. In addition, we compared the elapsed time of different algorithms on the MuJoCo dataset, as shown in Fig. 16.

## Funding

This work was supported by the National Natural Science Foundation of China (No. 62273356). The funders had no role in study design, data collection and analysis, decision to publish, or preparation of the manuscript.

## Grant Disclosures

The following grant information was disclosed by the authors:
The National Natural Science Foundation of China: 62273356.

## Competing Interests

The authors declare there are no competing interests.

## Author Contributions

- Tianlei Yao conceived and designed the experiments, performed the experiments, analyzed the data, performed the computation work, prepared figures and/or tables, authored or reviewed drafts of the article, and approved the final draft.
- Xiliang Chen conceived and designed the experiments, performed the computation work, authored or reviewed drafts of the article, and approved the final draft.
- Yi Yao conceived and designed the experiments, performed the computation work, authored or reviewed drafts of the article, and approved the final draft.
- Weiye Huang performed the experiments, analyzed the data, prepared figures and/or tables, and approved the final draft.
- Zhaoyang Chen performed the experiments, performed the computation work, prepared figures and/or tables, and approved the final draft.

## Data Availability

The public MuJoCo dataset is available at: https://rail.eecs.berkeley.edu/datasets/offline_rl/gym_mujoco_v2/

The Adroit dataset is available at https://sites.google.com/view/deeprl-dexterous-manipulation.

The Atari dataset is available at: https://console.cloud.google.com/storage/browser/atari-replay-dataset

The code is available at Github and Zenodo:

- https://github.com/NuvoleY/PLDT.git

- Yao, T. (2024). PLDT. Zenodo. https://doi.org/10.5281/zenodo.13558651.

## Supplemental Information

Supplemental information for this article can be found online at http://dx.doi.org/10.7717/peerj-cs.2490#supplemental-information.

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
