# Peer review of "Offline prompt reinforcement learning method based on feature extraction"

_PeerJ Computer Science, doi:10.7717/peerj-cs.2490_

## Round 0.1 · original submission · Major Revisions

After the first round of revision, “Major revision” is recommended based on the comments from the reviewers.

Authors should carefully address each of the comments raised by the reviewers and considering to implement the responses in the revised manuscript.

Reviewer 1 ·

Basic reporting

1) [Grammar] There are a few grammatical mistakes in the text:
a) In the abstract, change “we also divides the fixed-size state” to “we also divide the fixed-size state.. Extract the feature… encode”
b) Page 7 line 235, 236 -> "datasets" should be replaced by "dataset"
c) Line 303, missing space after comma
d) Line 154, missing space after colon

2) Some parts of the paper are a bit hard to understand during the first reading. For instance, the section on “Prompt Learning” (line 124) has scope for simplification. As an example, line 124 says "The goal of prefix information is to guide the model to extract information related to x, so as to better generate output information y", what is "x" supposed to represent here?

3) The inline references are not well formatted. For example, Page 3 line 124 has text such as “of Prefix TuningLi and Liang (2021)” and "templatesGu et al. (2022)” which are a bit hard to read, the authors should fix such cases.

4) Missing references: Page 10 mentions CQL algorithm but its unclear which paper in “References” section this maps to. I’d request authors to assign numbers to the papers in the “References'' section, and then correctly reference the text inline (example: CQL [20]) so it’s clear which reference belongs to which algorithm.

Experimental design

The investigations performed can be more rigorous. For instance, the paper currently uses Gym tasks from D4RL dataset (Halfcheetah, Hopper and Walker2d tasks), the authors can consider comparing the different methods across a wider range of tasks, for example the more complex Adroit tasks in the same D4RL dataset to understand how well the new method generalizes to diverse use-cases.

Validity of the findings

The authors can add a more thorough discussion about the failure cases of the proposed approach, and potential ways to mitigate such failures.

Additional comments

The text within figures starting Figure 5 is quite small and a bit hard to understand without zooming in, the authors should consider enhancing the quality of such images in the paper.

Cite this review as

Reviewer 2 ·

Basic reporting

The authors clearly identify the problem they're addressing - improving the stability and generalization of offline reinforcement learning methods. However, they could more explicitly state their research questions. The authors claim PLDT achieves better performance than baseline methods on several continuous control tasks from the benchmark, showing improved stability and generalization. However, the writing quality is not good enough. In addition, there are still have some problems need to improvement in this paper.
1. The paper does not provide a rigorous theoretical foundation for why the proposed method works better. A more in-depth analysis of the underlying principles would strengthen the work.
2. While the authors compare PLDT to some baseline methods, they do not include comparisons to other state-of-the-art offline RL algorithms. A more comprehensive comparison would better demonstrate the method's advantages.
3. The ablation experiments are limited to just two components of the method. More extensive ablations would help readers understand the importance of each part of the proposed approach.
4. All the formula lacks explanation, so it is hard to know the meaning of formula. Especially, for the formula parameters.
5.The figures to give the title is enough. The explanations of figures are better to put in the article.
6. The Pseudocode of algorithms A and B are hard to read. An algorithm need to have input, output and steps. The authors only define a function in algorithm A and only a for loop in the Algorithm B which pare poor for algorithms.
7.Clearly check the format of each reference paper format, such as "Prudencio, R. F., Maximo, M. R., and Colombini, E. L. (2022). A survey on offline reinforcement learning: Taxonomy, review, and open problems. IEEE transactions on neural networks and learning systems, 444 PP."

Experimental design

There are some suggestion for authors.
1.The paper does not provide detailed information on how the prompt trajectories are selected.
2. The experiments are primarily focused on MuJoCo continuous control tasks. Testing on a wider range of environments, including discrete action spaces, would better demonstrate the method's generalizability.
3. The paper does not discuss the computational requirements of PLDT compared to baseline methods. This information is important for practical applications.
4. The authors do not provide a comprehensive discussion of the potential limitations or failure cases of their method. A more balanced presentation of the approach's strengths and weaknesses would improve the paper.
5. What are the baseline method DT, CQL, BRAC, BC, StAR and GDT. Authors need to give full name and explain them.
6.The authors provide a high-level description of their method and some implementation details. However, more specific information on hyperparameters, computing resources used, and data preprocessing steps would be necessary for full replication.

Validity of the findings

Some suggestions:
1. What are the baseline method DT, CQL, BRAC, BC, StAR and GDT. Authors need to give full name and explain them.
2. The authors provide a high-level description of their method and some implementation details. However, more specific information on hyperparameters, computing resources used, and data preprocessing steps would be necessary for full replication.
3. The conclusion lack future work directions based on the findings and limitations.
4. Authors suggest to summarize main contributions of the paper. Emphasize how PLDT addresses the limitations of existing methods and its key innovations.

Additional comments

1. Authors did not cite Fig 1, 2, 3 in the paper and Table 1 is not cited in the paper, too.
2. The authors do not indicate the contribution of this paper clearly. It is better to give in the Introduction session.

Cite this review as

---

## Round 0.2 · accepted · Accept

Authors have addressed all the comments from the reviewers. Hence, this paper is recommended to accept in its current form.

Reviewer 1 ·

Basic reporting

The authors have addressed my prior comments in this section, the revised manuscript has better readability and some sections have been simplified as requested.

Experimental design

The authors have addressed my prior comments in this section, and have further refined the experiment as requested, adding tests under the Adroit task to better demonstrate the generalization of the method.

Validity of the findings

The authors have addressed my prior comments in this section, adding a section on failure cases and limitations of the method,

Additional comments

The authors have addressed my prior comments in this section.

Cite this review as

Reviewer 2 ·

Basic reporting

This is a revised paper. The authors have corrected their errors and modified their content.

Experimental design

This is a revised paper. The authors have enhanced their experimental design.

Validity of the findings

The authors have indicated their contributions in the revised version.

Additional comments

No additional comments

Cite this review as